# High-Frequency Ultrasound in Diagnosis and Treatment of Non-Melanoma Skin Cancer in the Head and Neck Region

**DOI:** 10.3390/diagnostics13051002

**Published:** 2023-03-06

**Authors:** Tiberiu Tamas, Cristian Dinu, Lavinia Manuela Lenghel, Emil Boțan, Adela Tamas, Sebastian Stoia, Daniel Corneliu Leucuta, Simion Bran, Florin Onisor, Grigore Băciuț, Gabriel Armencea, Mihaela Băciuț

**Affiliations:** 1Department of Maxillofacial Surgery and Implantology, Faculty of Dentistry, “Iuliu Hațieganu” University of Medicine and Pharmacy, 400012 Cluj-Napoca, Romania; 2Department of Radiology, “Iuliu Hatieganu” University of Medicine and Pharmacy, 400012 Cluj-Napoca, Romania; 3Department of Pathology, Emergency County Hospital, 400437 Cluj-Napoca, Romania; 4Doctoral School, “Iuliu Hatieganu” University of Medicine and Pharmacy, 400012 Cluj-Napoca, Romania; 5Medical Informatics and Biostatistics Department, “Iuliu Hatieganu” University of Medicine and Pharmacy, 400012 Cluj-Napoca, Romania

**Keywords:** non-melanoma skin cancer, basal cell carcinoma, squamous cell carcinoma, ultrasound, echography, high frequency transducer, skin, imaging

## Abstract

Non-melanoma skin cancer is one of the most frequently diagnosed cancers in the human body and unfortunately the incidence continues to increase. NMSC is represented by the basal cell carcinomas (BCCs) and squamous cell carcinomas (SCCs), which are the most prevalent forms, and basosquamous cell carcinomas (BSC) together with Merkel cell carcinoma (MCC), which are rare types but with a very aggressive pattern and poor prognosis. The pathological diagnosis is hard to assess without a biopsy, even by the dermoscopy. Moreover, the staging can be problematic because there is no access clinically to the thickness of the tumor and the depth of the invasion. The aim of this study was to evaluate the role of ultrasonography (US), which is a very efficient imaging method, non-irradiating and cheap, in diagnosis and treatment of non-melanoma skin cancer in the head and neck region. Thirty-one patients with highly suspicious malignant lesions of the head and neck skin were evaluated in the Oral and Maxillo-facial Surgery Department and Imaging Department in Cluj Napoca, Romania. All tumors were measured with three transducers: 13 MHz, 20 MHz and 40 MHz. Doppler examination and elastography were also used. The length, width, diameter, thickness, the presence of necrosis, status of regional lymph nodes, the presence of hyperechoic spots, strain ratio and vascularization were all recorded. After that, all patients were treated by surgical resection of the tumor and reconstruction of the defect. Immediately after surgical resection, all tumors were measured again after the same protocol. The resection margins were evaluated by all three types of transducers in order to detect malignant involvement and the results were compared with the histopathological report. We found that the 13 MHz transducers offered a big picture of the tumor but the level of details, in the form of the presence of the hyperechoic spots, is reduced. We recommend this transducer for evaluation of surgical margins or for the large skin tumors. The 20 and 40 MHz transducers are better for viewing the particularities of malignant lesions and for an accurate measurement; however, in the case of large size lesions, assessing all three dimensions of the tumor can be difficult. The intralesional hyperechoic spots are present in case of BCC and they can be used for differential diagnosis of BCC.

## 1. Introduction

One of the most common diagnosed tumors in the human body is non-melanoma skin cancer (NMSC). Unfortunately, according to GLOBOCAN, the incidence is steadily rising. Basal cell carcinomas (BCCs) and squamous cell carcinomas (SCCs) are the most prevalent forms, with a good prognosis in general, especially when they are found in the early stages. Other NMSC types are baso-squamous cell carcinomas (BSCs) and Merkel cell carcinomas (MCSs), with a more aggressive behavior and a poorer prognosis overall [1,2,3]. Because of the constant exposure to ultraviolet (UV) sun radiation, the head and neck region is the most affected anatomical area. Several alternative methods of treatment were documented for NMSC. Topical chemotherapy, cryotherapy, laser or thermocauterization, and superficial radiotherapy have demonstrated modest results. Surgical excision is still the treatment of choice, and achieving favorable cosmetic outcomes will always be difficult [4,5]. Clinical examination, palpation and dermoscopy are used to diagnose skin cancer, which is then followed by an incisional or excisional biopsy. Dermoscopy can be used to evaluate the lesion’s longitudinal and horizontal extension, but it cannot examine the lesion’s thickness or potential for invasion of adjacent tissues. The depth of invasion (DOI) from the American Joint Committee on Cancer (AJCC) staging system is used for melanoma and oral cavity cancer. In the last edition of the staging system, it was introduced for non-melanoma skin cancer and it modifies the stage when the invasion is deeper than 6 mm directly to T3. Additionally, DOI is correlated with regional metastasis in the case of SCC and other aggressive skin malignancies [6,7,8].

The imaging’s role is not very well defined. Magnetic resonance imaging (MRI) is a non-irradiating imaging method which is considered the gold standard for soft tissue tumors. Unfortunately, due to the small size of the tumors, MRI is not very efficient in this type of pathology. However, it can be a good option for very extensive tumors with local and regional adjacent structures involvement. Computed tomography scan (CT) involves the use of X-rays, and it cannot differentiate between soft tissues as well as the MRI. The advantage of this method is the ability to evaluate the bone involvement. It is reserved for very extensive tumors with high risk of bone tissue invasion [6,7,8,9].

Ultrasonography (US) is a very efficient imaging method, non-irradiating and cheap and, for these reasons, has a very high rate of acceptance from the patients. The thickness of the tumor, the depth of invasion, the invaded adjacent structures, the vascularity and the stiffness of the tumor may all be determined with an ultrasound examination. The disadvantage of US is the dependence on the examinator’s knowledge. The tumors should be evaluated by a trained practitioner in skin malignancy evaluation [6,10,11].

In recent years, new developments, such as high frequency transducers, have been introduced, as the structures of interest are only a few millimeters below the skin’s surface. These technologies can offer a broad range of data, including the precise anatomic position, shape, extent in all directions, blood flow and deeper layers involvement. Higher frequencies come with better resolution but less tissue penetration. In order to measure the thickness of the skin tumors, US transducers with a frequency range of 10 to 40 MHz are frequently utilized in clinical practice. Generally, NMSC are hypoechoic lesions with an inhomogeneous aspect on grayscale US examination [12,13]. In the prospective research by Song et al., high frequency US has shown positive outcomes in determining the DOI prior to surgery. To this end, the distance between the epidermal granulous layer and the deepest point of invasion is measured. An invasion deeper than 6 mm will place the tumor in stage III, even if the diameter of tumor is smaller than 2 cm. The upstaging of tumors means a poorer prognosis and a higher rate of positive margins after surgical treatment [14].

The technique of ultrasonography requires three methods for evaluation. The B-mode ultrasound may be followed by a color or power Doppler US scan of the lesion in order to obtain further details. All diameter measurements, including longitudinal, transverse and deep axe, should be recorded. The peak systolic velocity of the arterial flow, as well as information regarding the size and nature (arteries or veins) of the tumor vessels, are provided. Involvement of deeper layers, including cartilage, muscle and bone should be mentioned, if present. The form, depth, echogenicity (hypoechoic, isoechoic and hyperechoic), homogeneity (homogenous and heterogeneous), size, level of invasion and vascularity must all be included in the ultrasound report. An interesting aspect is the presence of the calcification in some malignant tumors which can be visualized as hyperechoic spots without acoustic shadow. Ulcerated lesions with hemorrhage and crusts can give artifacts and should be carefully evaluated [15,16]. The examinator should be familiarized with the manipulation of the transducers because the US beam is necessary to be perpendicular to the lesion and should find the largest diameter of the tumor. If too much pressure is applied on the tumor, the size can be easily modified and the staging and treatment plan compromised. A high amount of gel and low pressure should be used during the examination [17,18,19].

Strain elastography is a US examination which measures the stiffness of the tissues. The soft and hard tissues are color-coded differently on the strain elastogram. Usually, malignant tumors are stiffer than normal adjacent tissue, which makes them easily distinguishable. The images can be superposed and compared with those from grayscale examination [6,20].

It has also been suggested that optical coherence tomography (OCT), a high-resolution method based on the characteristics of light, can be used to analyze skin cancers. For thickness assessment in BCC lesions less than 2 mm in size, OCT appears to be more accurate and less biased than high-frequency US, according to the literature [21,22,23,24]. Other high-resolution imaging techniques include confocal microscopy, also known as reflectance confocal microscopy, and multiphoton microscopy, which may produce in vivo pictures that are almost as good as those from histologic examination of BCC lesions. In a histologic examination, a grayscale horizontal view of the tumor is displayed, which is in contrast to earlier cross-sectional imaging methods [6,25,26]. The differences between these techniques are summarized in Table 1.

The gold standard for NMSC treatment is surgical excision and defect reconstruction. Other treatments can be attempted, such as local chemotherapy for very small lesions, or systemic chemotherapy, immunotherapy, Hedgehog Pathway inhibitors, and radiotherapy for highly advanced tumors. In order to attain good quality surgery with negative resection margins, it is necessary to know the diameter and the thickness of the tumor preoperatively. Because the thickness is not accessible without imaging, Mohs surgery should be attempted if it is available. If not, delaying the defect reconstruction until the certification of complete resection is also an option.

The aim of this study is to evaluate the role of ultrasonography in diagnosis, staging and treatment of the NMSC of the head and neck region.

## 2. Materials and Methods

Thirty-one patients with highly suspicious malignant lesions of the head and neck skin were evaluated in the Oral and Maxillo-facial Surgery Department and Imaging Department in Cluj Napoca, Romania. All patients were informed of the study and agreed to participate. The Ethical Committee of the Emergency County Hospital Cluj-Napoca, Romania (39109/A/0001/UK/R Reg. n. 362/08.01.2020) and the Iuliu Hatieganu University of Medicine and Pharmacy Cluj-Napoca, Romania (433/11.10.2019) approved the study. After clinical examination, the patients were investigated by ultrasonography with high frequency transducers. All tumors were measured with three transducers: 13 MHz, 20 MHz and 40 MHz (Figure 1). Doppler examination and elastography were also used. The length, width, diameter, thickness, the presence of necrosis, status of regional lymph nodes, the presence of hyperechoic spots, strain ratio and vascularization were all recorded. All patients were treated by surgical resection of the tumor and reconstruction of the defect (Figure 2). Immediately after surgical resection, all tumors were placed with the epidermis facing up on sterile gauze and fixed with small-gauge needles. A layer of US gel was applied. The tumors were measured again after the same protocol. Doppler and elastography techniques were not performed. The resection margins were evaluated by all three types of transducers in order to detect malignant involvement of the resection margins or to measure the closest margin to the tumor. The results were compared with the histopathological report.

### Statistical Analyses

To choose the interclass correlation coefficient (ICC), we followed the guide of Koo et al. For the type of reliability study, we considered an inter-rater reliability study, with having the same set of raters for all subjects that were not randomized; thus, a two-way mixed effects analysis was chosen. The intended measurement protocol was based on a single rater, and we considered consistency to be the most important quality. Despite that, we did also compute the absolute agreement. We used the interpretation of Portney et al. for ICC: 0.5 suggesting poor reliability, values between 0.5 and 0.75 suggesting moderate reliability, values between 0.75 and 0.9 suggesting good reliability and values greater than 0.90 suggesting excellent reliability. For ICC computations, we used the irr R package [27,28,29].

Concerning the concordance between the echographic diagnosis, or characteristics (as hyperechoic spots) with the histopathological diagnosis, the inter-rater agreement was sought, as evidenced by Cohen Kappa (following Altman’s interpretation: 0.20—poor; 0.21–0.40—fair; 0.41–0.60—moderate; 0.61–0.80—good; 0.81–1.00—very good). Kappa can vary from -1 (lower agreement than expected by chance) to 1 (higher agreement than expected) and is the norm of the difference between the actual rate of agreement and the chance rate of agreement anticipated (almost perfect agreement). However, the kappa coefficient does not account for inter-observer variability (degree of disagreement) or data prevalence (how evenly categories are distributed); thus, we computed the prevalence-adjusted, bias-adjusted Cohen Kappa [30,31,32].

## 3. Results

Concerning concordance between the histopathological and echographic measurements of tumor thickness, we found excellent agreement for all transducers, with the best ones being for the preoperative observations and the least favorable ones being the postoperative observations (Figure 3) (Table 2). Within any of the preoperative and postoperative observations, the 40 MHz transducer fared the best, followed by the 20 MHz transducer and finally by the 13 MHz transducer.

We assessed the concordance between the final diagnosis and ultrasound diagnosis, and we found a Cohen’s Kappa of 0.8897 (95% CI: 0.6783–1.1011), bias-adjusted Kappa of 0.8895 and prevalence- and bias-adjusted Kappa of 0.9355 (Table 3). The concordance between the final diagnosis of BCC and the presence of hyperechoic spots followed an identical pattern. We have associated the presence of the hyperechoic spots with diagnosis of BCC (Figure 4). Only one case of BCC was misdiagnosed by echography due to the missing hyperechoic spots (Figure 5).

Regarding elastography, we can easily see that the tumor tissue is stiffer than the normal adjacent one (Figure 6). We did not find any significant statistical association between the value of strain ratio and the histology of the tumor. The values in the BCC group were also lower than in the SCC group (95% CI −2.98–0.35, *p* = 0.126). Doppler examination (Figure 7) showed increased values for both systolic and diastolic speed in the SCC group, but the results were not statistically significant and no association between the type of histology and vascularization was found (Table 4). 

We found a very high correlation between the ultrasound measurement of the surgical margins and pathological reports, with the best result being achieved with the 13 MHz transducer, followed by the 20 MHz and 40 MHz transducers (Figure 8) (Table 5).

## 4. Discussion

The skin is defined as the largest organ in the human body and it is also the first line of defense from different external aggressors. This protective line can be harmed by environmental factors, including ultraviolet radiation or chemicals, leading to the development of skin cancer. The incidence of both melanoma and non-melanoma malignancies is rapidly increasing on a worldwide scale. According to GLOBOCAN statistics data, there were more than 1 million new cases globally in 2020, placing NMSC fourth among all malignancies. The death rate is reduced when compared to melanoma or other cancers. There are various subtypes, including Merkel cell carcinoma, cutaneous squamous cell carcinoma, basal cell carcinoma and basosquamous cell carcinoma. There are also different types of presentation among the same histological type as nodular BCC, infiltrative BCC, Morpheus type BCC, etc. There are few studies in the literature that discuss the function of the US in diagnosing, staging and treating of NMSC, particularly in the head and neck area [33,34]. 

Staging cancer means to start with measurement and assessment of all three dimensions of the tumor (length, width and depth). The updated AJCC staging system uses the invasion measurement in staging of squamous cell carcinoma. In addition, the risk of regional metastases in proportion with the depth of invasion for the aggressive skin cancers as SCC, BSC or MCC [33]. Even if some clinicians choose the biopsy for diagnosis and staging of the highly suspicious lesions, this method can lead to staging errors because the thickness of the tumor is unknown before surgical intervention. This means that the chance to have positive surgical margins and an incorrect staging are higher. Dermoscopy can help the diagnosis process but not the staging process because the thickness of the tumor is not measurable by this method. 

The US transducers frequency used in other studies varies between 5–18 MHz. We used three different transducers in order to obtain the closest measurement and to see the advantages and disadvantages of each one. The 13 MHz transducers offered a big picture of the tumor but the level of details, in the form of the presence of the hyperechoic spots, is reduced. We recommend this transducer for evaluation of surgical margins or for large skin tumors. The 20 and 40 MHz transducers are better for viewing the particularities of malignant lesions and for an accurate measurement but in the case of large size lesions, assessing of all three dimensions of the tumor can be difficult. Increasing the resolution of transducers obtains better level of details but, unfortunately, the level of penetrance decreases [6]. 

Dermoscopy can help the preoperative diagnosis process. Pampena R. et al. have examined retrospectively more than 400 dermoscopy images in order to find if it is possible to differentiate between the types of BCC. They found a prevalence of arborizing telangiectasia in invasive BCC (iBCC), as compared to superficial BCC (sBCC), but there were no significant differences in arborizing telangiectasia between iBCC and nodular BCC (nBCC). They found ulcerations more frequently in iBCC than both in sBCC and nBCC [34]. Because high frequency ultrasound and Doppler examination can evaluate both the vascularization and the presence of ulcerations, its use should be combined with dermoscopy in order to strengthen the results in BCC type differentiation. Knowing that the treatment should be more aggressive for SCC, with wider margins of resection and in some cases completed with neck dissection and even superficial parotidectomy, it is very important to be sure about the preoperative diagnosis. 

The incisional biopsy is done for the big lesions but is very difficult or impossible to achieve in the case of small lesions. For these lesions, an excisional biopsy with wide margins is used for diagnosis and also for treatment. Mohs surgery can help for diagnosis and proper treatment of skin cancer. On ultrasound, the BCC and SCC look quite similar, except the BCC presents some hyperechoic spots. They look like a cotton flower but the particularity is the absence of the acoustic shadowing [35]. The spots are histologically represented by microcalcifications, horn cysts and apoptotic cells. Alfageme Roldan F has studied these spots and he found two interesting aspects: the increased number of spots is associated with an aggressive behavior and the micronodular type presents the highest number of spots. These results could be very useful in the diagnosis process and treatment plan [36]. In accordance with the other studies, we found an excellent correlation between the presence of hyperechoic spots and the diagnosis of BCC. We recommend this method for differentiating the BCC from other skin malignancies and for avoiding extending the surgical excision unnecessarily.

The strain value in strain elastography serves as a proximate indicator of tissue stiffness or elasticity. Variable skin layers have different elastic qualities, with subcutaneous tissue being more elastic and the epidermis and dermis being less elastic. Increased marginal stiffness is very suggesting for cancer. The malignant tissue develops a fibrotic environment that makes it stiffer than the surrounding tissues [36,37]. Some researchers came to the conclusion that elastography is a variable evaluation that, when combined with B-mode results, may provide the physician or the surgeon with important knowledge regarding different subtypes of BCC. More specifically, infiltrative and non-infiltrative cutaneous basal cell carcinomas are distinguished by enhanced marginal stiffness [38]. Tanaka et al. showed that the tumor thickness of nodular BCC determined by elastography corresponded better with the histological thickness than the tumor thickness measured by traditional B-mode sonography. These results suggest that elastography can be a dependable approach for planning future surgical safety margins since the infiltrative variety of BCC carries a greater risk of recurrence [39]. We did not find any correlation between the nodular or infiltrative type of NMSC and the stiffness of the tumor. We also did not try to measure the thickness of the tumor on elastography, so we cannot confirm the previous result. A certainty that we observed is that the increased stiffness of the tumor compared with the normal adjacent one, which can help us establish the diagnosis of malignant tumor and guide us in surgical treatment.

The treatment of NMSC is based on surgical excision, regional lymph nodes excision if necessary, and other forms of therapy such as radiation therapy, chemotherapy or immunotherapy in advanced cases. NCCN recommendations are based on a large study by Wolf and Zitelli. This study involved 117 cases of BCCs. The tumors were divided regarding the diameter: greater than 2 cm and smaller than 2 cm. They performed the surgical excision at 4 mm around the tumors. For tumors smaller than 2 cm, the complete removal was obtained in 95% of all cases. For the other tumors, the rate of complete removal was lower and they recommended larger margins than 4 mm [40].

The European Dermatology Forum (EDF) guidelines regarding the surgical margins for BCC are based on the previous EDF guidelines, the British Association of Dermatology guidelines (BAD), and the French guidelines. The recommendations are close to NCCN guidelines. If the BCC is low-risk (less than 2 cm in diameter) the surgical margins can be placed at 3–4 mm around the tumor. They also do not recommend doing the same for larger tumors because the risk of positive margins is higher, so they suggest 5 to 10 mm surgical margins. They confirm the results of previous studies with a complete excision rate of 95% when the mentioned protocol was used [41,42]. Because these tissues are considered natural barriers, the surgical excision should be conducted to the level of fascia perichondrium or periosteum when possible. If the tumor is located in a region with thicker skin, the excision can be done more superficially. Moreover, the type of BCC should guide the surgical excision because the baso-squamous cell carcinoma and morpheus type BCC have a poorer outcome [41,42].

Because a high percentage of the defects are closed immediately with local and regional flaps, finding positive margins based on a pathological report can be problematic. Mohs surgery, named after Frederick Mohs, the surgeon who invented it, is a technique where the surgical margins are microscopically controlled during the surgical intervention. This surgical method provides high cure rates in the treatment of a range of skin malignancies. Unfortunately, not all medical institutions provide it. Evaluation of margin status prior to a histology report would be nearly impossible without this approach [43].

Regarding the role of US in the treatment of NMSC, few studies concluded that US can easily detect malignant involvement of the margins. This may play a role in the treatment of skin cancer and prevention of local recurrence, providing improved overall survival rates. The adjacent normal tissue is measured from the limit of the tumor to the surgical resection margin. It can be easily measured on grayscale examination or elastography [44,45]. Our findings regarding the evaluation of surgical margins postoperatively illustrate the utility of this technique for immediate measurement of surgical margins directly in the operating room. All three transducers (13, 20 and 40 MHz) have shown very good results, but the most accurate was the 13 MHz transducer. This can be explained by how the lower the frequency, the deeper or greater penetrance a sound wave has. If positive or close margins are visualized, the surgery can be completed immediately. This way, the surgeon can avoid insufficient resection of the tumor or unnecessary sacrifice of healthy tissue. All these should reduce the recurrence rate and improve overall survival, but further studies are necessary to ascertain this fact.

Last but not least, US can be very useful for diagnosis and staging of melanoma skin cancer, which is still one of the deadliest types of cancer. A very precise DOI should be measured because the staging and the treatment change from a difference of just 1 mm. The treatment of both melanoma and non-melanoma skin cancer is difficult in the head and neck region due to the multiple anatomical vital elements, and the large excisions recommended for other anatomical regions are frequently not applicable. Comparative studies between these two pathologies could be further directions for skin cancer research.

## 5. Conclusions

NMSC is one of the most frequent malignancies in the human body and the incidence is quickly rising. The treatment of these tumors is mainly surgical but can differ for BCC, SCC and other types. For this reason, an appropriate preoperative diagnosis is necessary. Clinical examination and dermoscopy can help the diagnosis process, but only the pathological report can offer the final diagnosis. Unfortunately, for small tumors, only excisional biopsy can be done, which is primarily performed at the same time as surgical treatment of the lesion. Ultrasound examination with high frequency transducers can differentiate the BCC from other types of skin malignancy and is very easily accepted by the patient because it is a non-irradiating and non-invasive technique. Unnecessary healthy tissue sacrifice can be avoided with this technique. The staging of the tumor has the same importance as the diagnosis. Dermoscopy and clinical examination cannot evaluate the thickness of the tumor, which is very important for treatment and also for prognosis. Echography with high frequency transducers and elastography can measure precisely the thickness of the tumor and can guide the tumor excision, avoiding the tumor involvement of the surgical margins and reducing the recurrence rate. Involvement of positive margins and detection of close margins can be easily detected with 13, 20 and 40 MHz transducers.

## Figures and Tables

**Figure 1 diagnostics-13-01002-f001:**
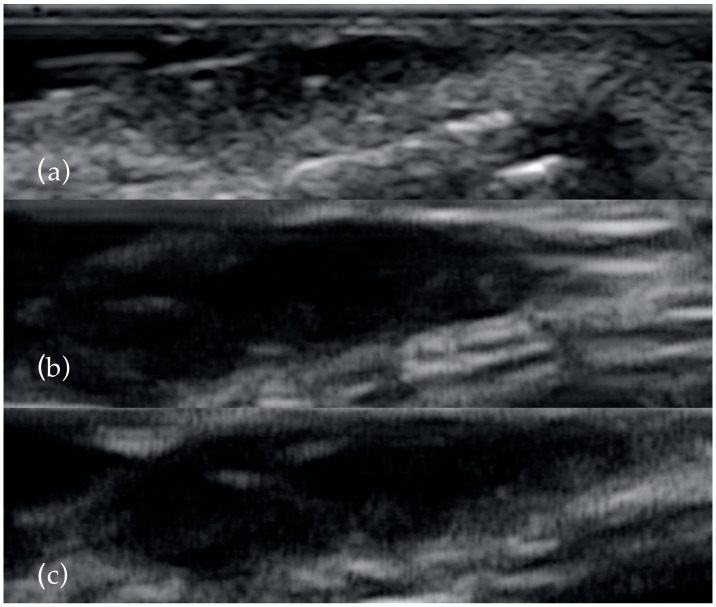
The same BCC examined with 13 MHz (**a**), 20 MHz (**b**) and 40 MHz (**c**). As the frequency increases, the presence of the two hyperechoyc spots is clearer.

**Figure 2 diagnostics-13-01002-f002:**
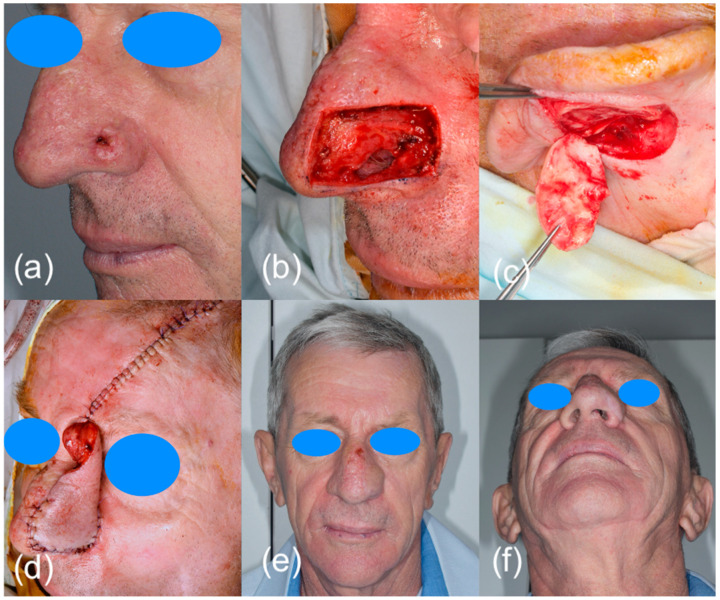
Surgical excision of a nasal BCC: (**a**) clinical presentation of the malignant lesion; (**b**) surgical defect after tumor resection together with alar cartilage and a portion of nasal mucosa; (**c**) auricular cartilage graft for prefabricated frontal flap; (**d**) reconstruction of the nose with prefabricated frontal flap and auricular cartilage; (**e**,**f**) clinical aspect 1 month after surgery.

**Figure 3 diagnostics-13-01002-f003:**
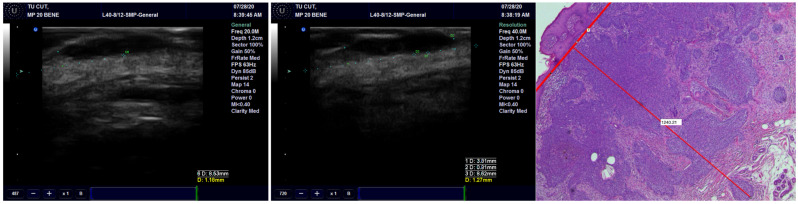
The thickness of the tumor, measured on US with 20 MHz and 40 MHz transducers, compared with the pathological image of the tumor showing very high occurrence of US.

**Figure 4 diagnostics-13-01002-f004:**
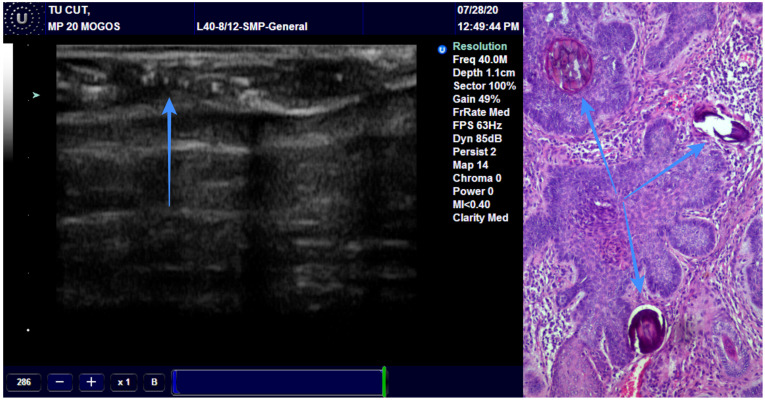
The arrow is showing the presence of the hyperechoic spots in a BCC and the pathological correspondence.

**Figure 5 diagnostics-13-01002-f005:**
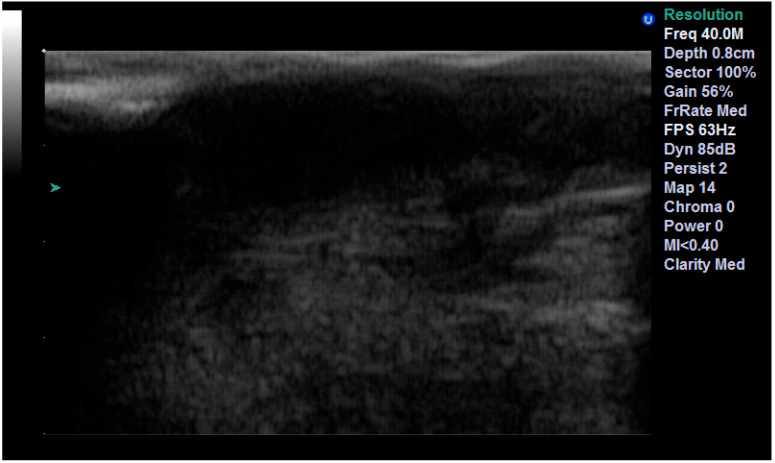
Misdiagnosed BCC due to the absence of hyperechoic spots.

**Figure 6 diagnostics-13-01002-f006:**
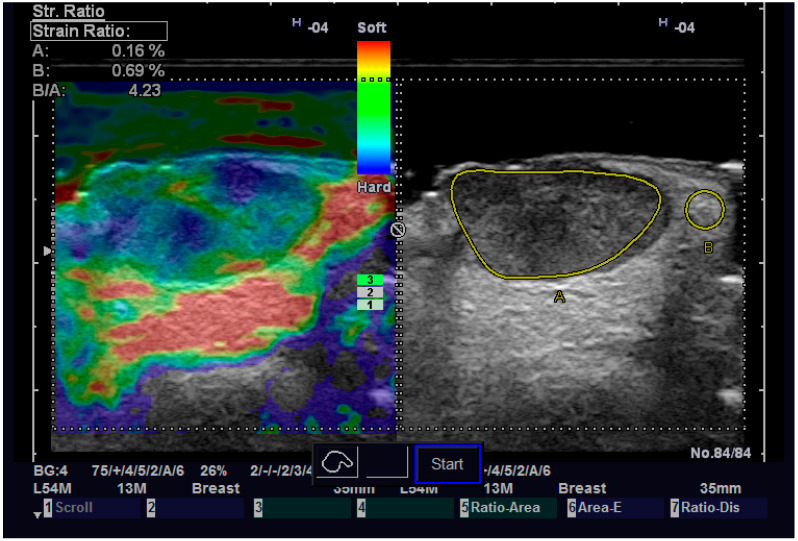
Strain elastography of a SCC.

**Figure 7 diagnostics-13-01002-f007:**
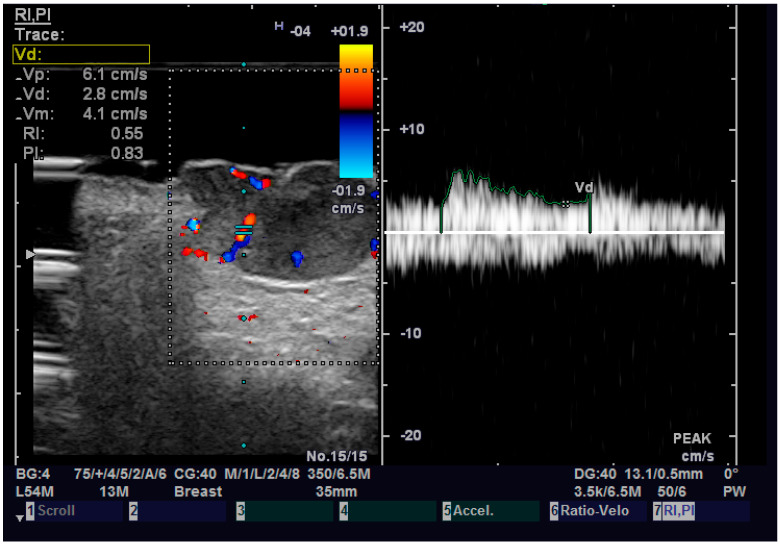
Doppler examination of a SCC.

**Figure 8 diagnostics-13-01002-f008:**
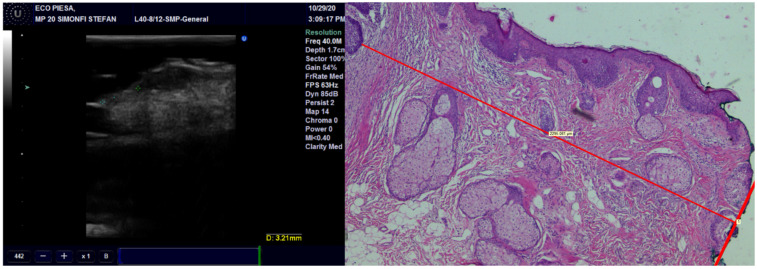
Surgical margins measurement by US and histopathology.

**Table 1 diagnostics-13-01002-t001:** Characteristics of different non-invasive tehniques used for evaluation of NMSC.

	Ultrasonography	Dermoscopy	OCT	Confocal Microscopy
Assessing the size of the tumor	It can assess the size in all three dimensions including the thickness [14]	It can assess the size except the thickness [1,6]	It can assess the size in all three dimensions for tumors with a thickness less than 2 mm [21,22,23,24]	It can assess the size in all three dimensions for tumors with a thickness less than 3 mm [25,26]
Type of subjacent tissue involvement	It can precisely identify the type of subjacent tissue involved (muscle, fascia, perichondrium)	It cannot precisely identify the type of subjacent tissue involved	Because the depth of penetration is less than 2 mm, we do not recommend using it for evaluation of subjacent tissue involvement	Because the depth of penetration is less than 3 mm, we do not recommend to using it for evaluation of subjacent tissue involvement
Surgical margins involvement detection	Yes (postoperative evaluation of the excised specimen)	Only preoperatively	Only preoperatively. Very limited data in the literature	Yes (Ex vivo confocal microscopy)
Vascularity	Yes	Yes	Yes (Dynamic Optical Coherence Tomography)	Yes
Specificity is diagnosis	85–90% for both BCC and SCC	95–99% for BCC Insufficient data for SCC	80–100% for SCC 70–80% for BCC	92–98% for SCC 93% for BCC
Sensitivity is diagnosis	93–95% for both BCC and SCC	90–93% for BCC 75–77% for SCC	92–93% for SCC 92–95% for BCC	74–77% for SCC 92% for BCC

**Table 2 diagnostics-13-01002-t002:** Tumor thickness measured pre- and post-operatively, with 13, 20 and 40 MHz transducers, as well as the inter-class correlation coefficient between echography and pathological measurements.

Method	Tumor Thickness (mm), Median (IQR)	Difference (95% CI)	*p*-Value	ICC Consistency (95% CI)	*p*-Value	ICC Agreement (95% CI)	*p*-Value
Pathology	2.5 (1.98–4.5)						
Preoperative							
13 MHz	2.5 (1.75–4.4)	0 (0–0.32)	0.063	0.982 (0.962–0.991)	<0.001	0.98 (0.957–0.99)	<0.001
20 MHz	2.3 (1.75–4.4)	0.2 (−0.09–0.18)	0.574	0.981 (0.96–0.991)	<0.001	0.981 (0.961–0.991)	<0.001
40 MHz	2.2 (1.9–4.5)	0.3 (−0.07–0.15)	0.394	0.992 (0.984–0.996)	<0.001	0.992 (0.985–0.996)	<0.001
Postoperative							
13 MHz	2.3 (1.45–4)	0.2 (0.13–0.65)	0.008	0.927 (0.853–0.964)	<0.001	0.909 (0.776–0.96)	<0.001
20 MHz	2.35 (1.6–4.05)	0.15 (0.1–0.6)	0.011	0.927 (0.855–0.964)	<0.001	0.911 (0.787–0.96)	<0.001
40 MHz	2.4 (1.6–3.95)	0.1 (0.07–0.58)	0.012	0.93 (0.859–0.965)	<0.001	0.916 (0.806–0.962)	<0.001

**Table 3 diagnostics-13-01002-t003:** Concordance between histopathological diagnosis, echographic diagnosis and hyperechoic spots.

Histopathological Diagnosis	Basal Cell Carcinoma	Squamous Cell Carcinoma	Total
Echographic diagnosis			
Basal cell carcinoma	25	0	25
Squamous carcinoma	1	5	6
Total	26	5	31
Hyperechoic spots			
Present	25	0	25
Absent	1	5	6
Total	26	5	31

**Table 4 diagnostics-13-01002-t004:** Pulsatility index, resistive index, strain ratio, diastolic speed and systolic speed related to histopathological type.

Diagnostic Histopathologic	Cc Basal cell (n = 25)	Cc Squamous (n = 5)	Difference (95% CI)	*p*
Pulsatility index, median (IQR)	0.75 (0.6–0.94)	0.82 (0.54–0.95)	0.07 (−0.3–0.31)	0.889^●^[n1 = 25, n2 = 5]
Resistive index, median (IQR)	0.5 (0.4–0.6)	0.52 (0.42–0.63)	0.02 (−0.13–0.18)	0.933^●^[n1 = 25, n2 = 5]
Strain ratio, median (IQR)	1.94 (1.38–2.62)	3.2 (2.9–3.6)	1.26 (−2.98–0.35)	0.126^●^[n1 = 26, n2 = 5]
Diastolic speed, median (IQR)	3.9 (3.9–4.1)	4.3 (2.5–4.9)	0.4 (−1–1.6)	0.822^●^[n1 = 25, n2 = 5]
Systolic speed, median (IQR)	6.3 (6.1–9)	6.8 (6.4–8.2)	0.5 (−1.9–2.2)	0.758^●^[n1 = 25, n2 = 5]

**Table 5 diagnostics-13-01002-t005:** Correlation between the ultrasound measurement of the surgical margins and pathological reports.

Method	Tumor Margin (mm), Median (IQR)	Difference (95% CI)	*p*-Value	ICC Consistency (95% CI)	*p*-Value	ICC Agreement (95% CI)	*p*-Value
Pathology	2 (1–2.75)						
13 MHz	2 (1–2.65)	0 (−0.05–0.15)	0.367	0.993 (0.985–0.997)	<0.001	0.993 (0.985–0.997)	<0.001
20 MHz	1.9 (1.15–3)	−0.1 (−0.05–0.25)	0.203	0.961 (0.921–0.981)	<0.001	0.96 (0.92–0.981)	<0.001
40 MHz	2 (1.1–2.8)	0 (−0.05–0.27)	0.195	0.972 (0.942–0.986)	<0.001	0.971 (0.941–0.986)	<0.001

## Data Availability

The data presented in this study are available in the article.

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
