# Peer review of "High-Frequency Ultrasound in Diagnosis and Treatment of Non-Melanoma Skin Cancer in the Head and Neck Region"

_diagnostics, 2023, doi:10.3390/diagnostics13051002_

Round 1

Reviewer 1 Report

The authors report a study about high-frequency ultrasound in diagnosis and treatment of non-melanoma skin cancer in the head and neck region. The article is of interest for publication. Just some changes are needed:

Generally, explain better and more in detail which are the main differences bewtee ultrasound and other non invasive tecniques for the detection of head/neck non melanoma skin cancers.

In this regard, please add a table where you summarize the main differences of the current non -invasive methods for the detection of non melona skin cancers, such as confocal microscopy, dermoscopy, OCT

Several sentences in the manjuscript are twisted and there are several grammal errors, therefore please improve the English

In the Discussion, please add some sentence about the fact that this method may help for the detcetion and/or followup of invasive histotypes of basal cell carcinoma. In this regard please read this recent article and add in you references (Clinical and Dermoscopic Factors for the Identification of Aggressive Histologic Subtypes of Basal Cell Carcinoma. Front Oncol. 2021 Feb 19;10:630458. doi: 10.3389/fonc.2020.630458. PMID: 33680953; PMCID: PMC7933517.)

Please since the head/neck region is composed by facial aesthetic units, in studying, excising and evaluationg the tumors in these anatiomic regions, please add some sentence about the role of ultrasound in tumors (such as non melanoma skin cancers and melanoma). 

Author Response

Dear reviewer,

Thank you for the quick response and the valuable suggestions. As you have recommended, a table where the characteristics of non invasive methods of investigation are presented, is now introduced in the main text. I also read the article that you have recommended and I introduced the information and the citation in the text. Also a paragraph referring to both melanoma and non melanoma skin cancer was added.

Thank you again for all your support and for the quick response.

Kind regards,

Tiberiu Tamas

Reviewer 2 Report

I found the work very interesting and well presented.

But it would be useful to add some images showing the same lesion in the three frequencies ( three transducers, 13MHz, 20 MHz and 40 MHz.)

It would also be interesting to view the image extrapolated from  the one and only case of BCC that was misdiagnosed ( by echography due to the missing hyperechoic spots n-204 ) and the reasons for the misdiagnosis.

Author Response

Dear reviewer,

Thank you for your quick response and for the valuable suggestions. As you recommended, I have introduced in the main text an image with the evaluation of the same tumor with the three mentioned transducers (figure 1). I have also noted that the spots are clearer when the frequency increases. I have added the photo with the misdiagnosed BCC due to the absence of hyperechoic spots. We have explained that the diagnosis of BCC was established only in the presence of these spot. Because here the spots were absents, we unfortunately misdiagnosed the tumor. 

Thank you again for your help and for the quick response.

Kind regards,

Tiberiu Tamas

Round 2

Reviewer 1 Report

The Authors performed the changes and the article can be accepted for publication. Thank you.